# The Power of Optimization from Samples

**Eric Balkanski**
Harvard University
`ericbalkanski@g.harvard.edu`

**Aviad Rubinstein**
University of California, Berkeley
`aviad@eecs.berkeley.edu`

**Yaron Singer**
Harvard University
`yaron@seas.harvard.edu`

## Abstract

We consider the problem of optimization from samples of monotone submodular functions with bounded curvature. In numerous applications, the function optimized is not known a priori, but instead learned from data. What are the guarantees we have when optimizing functions from sampled data?

In this paper we show that for any monotone submodular function with curvature $c$ there is a $(1 - c)/(1 + c - c^2)$ approximation algorithm for maximization under cardinality constraints when polynomially-many samples are drawn from the uniform distribution over feasible sets. Moreover, we show that this algorithm is optimal. That is, for any $c < 1$, there exists a submodular function with curvature $c$ for which no algorithm can achieve a better approximation. The curvature assumption is crucial as for general monotone submodular functions no algorithm can obtain a constant-factor approximation for maximization under a cardinality constraint when observing polynomially-many samples drawn from any distribution over feasible sets, even when the function is statistically learnable.

## 1   Introduction

Traditionally, machine learning is concerned with *predictions*: assuming data is generated from some model, the goal is to predict the behavior of the model on data similar to that observed. In many cases however, we harness machine learning to make *decisions*: given observations from a model the goal is to find its optimum, rather than predict its behavior. Some examples include:

- **Ranking in information retrieval:** In *ranking* the goal is to select $k \in \mathbb{N}$ documents that are most relevant for a given query. The underlying model is a function which maps a set of documents and a given query to its relevance score. Typically we do not to have access to the scoring function, and thus learn it from data. In the *learning to rank* framework, for example, the input consists of observations of document-query pairs and their relevance score. The goal is to construct a scoring function of query-document pairs so that given a query we can decide on the $k$ most relevant documents.

- **Optimal tagging:** The problem of *optimal tagging* consists of picking $k$ tags for some new content to maximize incoming traffic. The model is a function which captures the way in which users navigate through content given their tags. Since the algorithm designer cannot know the behavior of every online user, the model is learned from observations on user navigation in order to make a decision on which $k$ tags maximize incoming traffic.

- **Influence in networks:** In *influence maximization* the goal is to identify a subset of individuals who can spread information in a manner that generates a large cascade. The underlying

assumption is that there is a model of influence that governs the way in which individuals forward information from one to another. Since the model of influence is not known, it is learned from data. The observed data is pairs of a subset of nodes who initiated a cascade and the total number of individuals influenced. The decision is the optimal set of influencers.

In the interest of maintaining theoretical guarantees on the decisions, we often assume that the generative model has some structure which is amenable to optimization. When the decision variables are discrete quantities a natural structure for the model is *submodularity*. A function $f : 2^N \to \mathbb{R}$ defined over a ground set $N = \{e_1, \ldots, e_n\}$ of elements is submodular if it exhibits a diminishing marginal returns property, i.e., $f_S(e) \geq f_T(e)$ for all sets $S \subseteq T \subseteq N$ and element $e \notin T$ where $f_S(e) = f(S \cup e) - f(S)$ is the *marginal contribution* of element $e$ to set $S \subseteq N$. This diminishing returns property encapsulates numerous applications in machine learning and data mining and is particularly appealing due to its theoretical guarantees on optimization (see related work below).

The guarantees on optimization of submodular functions apply to the case in which the algorithm designer has access to some succinct description of the function, or alternatively some idealized value oracle which allows querying for function values of any given set. In numerous settings such as in the above examples, we do not have access to the function or its value oracle, but rather learn the function from observed data. If the function learned from data is submodular we can optimize it and obtain a solution with provable guarantees on the learned model. But how do the guarantees of this solution on the learned model relate to its guarantees on the generative model? If we obtain an approximate optimum on the learned model which turns out to be far from the optimum of the submodular function we aim to optimize, the provable guarantees at hand do not apply.

**Optimization from samples.** For concreteness, suppose that the generative model is a monotone submodular function $f : 2^N \to \mathbb{R}$ and we wish to find a solution to $\max_{S:|S| \leq k} f(S)$. To formalize the concept of observations in standard learning-theoretic terms, we can assume that we observe samples of sets drawn from some distribution $\mathcal{D}$ and their function values, i.e. $\{(S_i, f(S_i))\}_{i=1}^m$. In terms of learnability, under some assumptions about the distribution and the function, submodular functions are statistically learnable (see discussion about `PMAC` learnability). In terms of approximation guarantees for optimization, a simple greedy algorithm obtains a $1 - 1/e$-approximation.

Recent work shows that optimization from samples is generally impossible [4], even for models that are learnable and optimizable. In particular, even for maximizing coverage functions, which are a special case of submodular functions and widely used in practice, no algorithm can obtain a constant factor approximation using fewer than exponentially many samples of feasible solutions drawn from any distribution. In practice however, the functions we aim to optimize may be better behaved.

An important property of submodular functions that has been heavily explored recently is that of *curvature*. Informally, the curvature is a measure of how far the function is to being *modular*. A function $f$ is modular if $f(S) = \sum_{e \in S} f(e)$, and has curvature $c \in [0, 1]$ if $f_S(e) \geq (1 - c)f(e)$ for any $S \subseteq N$. Curvature plays an important role since the hard instances of submodular optimization often occur only when the curvature is unbounded, i.e., $c$ close to 1. The hardness results for optimization from samples are no different, and apply when the curvature is unbounded.

*What are the guarantees for optimization from samples of submodular functions with bounded curvature?*

In this paper we study the power of optimization from samples when the curvature is bounded. Our main result shows that for any monotone submodular function with curvature $c$ there is an algorithm which observes polynomially-many samples from the uniform distribution over feasible sets and obtains an approximation ratio of $(1 - c)/(1 + c - c^2) - o(1)$. Furthermore, we show that this bound is tight. For any $c < 1$, there exist monotone submodular functions with curvature $c$ for which no algorithm can obtain an approximation better than $(1 - c)/(1 + c - c^2) + o(1)$ given polynomially many samples. We also perform experiments on synthetic hard instances of monotone submodular functions that convey some interpretation of our results.

For the case of modular functions a $1 - o(1)$ algorithm can be obtained and as a consequence leads to a $(1 - c)^2$ algorithm for submodular functions with bounded curvature [4]. The goal of this work is to exploit the curvature property to obtain the optimal algorithm for optimization from samples.

**A high-level overview of the techniques.** The algorithm estimates the expected marginal contribution of each element to a random set. It then returns the (approximately) best set between the set of elements with the highest estimates and a random set. The curvature property is used to bound the differences between the marginal contribution of each element to: (1) a random set, (2) the set of elements with highest (estimated) marginal contributions to a random set, and (3) the optimal set. A key observation in the analysis is that if the difference between (1) and (3) is large, then a random set has large value (in expectation).

To obtain our matching inapproximability result, we construct an instance where, after viewing polynomially many samples, the elements of the optimal set cannot be distinguished from a much larger set of elements that have high marginal contribution to a random set, but low marginal contribution when combined with each other. The main challenge is constructing the optimal elements such that they have lower marginal contribution to a random set than to the other optimal elements. This requires carefully defining the way different types of elements interact with each other, while maintaining the global properties of monotonicity, submodularity, and bounded curvature.

## 1.1 Related work

**Submodular maximization.** In the traditional value oracle model, an algorithm may adaptively query polynomially many sets $S_i$ and obtain via a black-box their values $f(S_i)$. It is well known that in this model, the greedy algorithm obtains a $1 - 1/e$ approximation for a wide range of constraints including cardinality constraints [23], and that no algorithm can do better [6]. Submodular optimization is an essential tool for problems in machine learning and data mining such as sensor placement [20, 12], information retrieval [28, 14], optimal tagging [24], influence maximization [19, 13], information summarization [21, 22], and vision [17, 18].

**Learning.** A recent line of work focuses on learning submodular functions from samples [3, 8, 2, 10, 11, 1, 9]. The standard model to learn submodular functions is $\alpha$-PMAC learnability introduced by Balcan and Harvey [3] which generalizes the well known PAC learnability framework from Valiant [26]. Informally, a function is PAC or PMAC learnable if given polynomially samples, it is possible to construct a function that is likely to mimic the function for which the samples are coming form. Monotone submodular functions are $\alpha$-PMAC learnable from samples coming from a product distribution for some constant $\alpha$ and under some assumptions [3].

**Curvature.** In the value oracle model, the greedy algorithm is a $(1 - e^{-c})/c$ approximation algorithm for cardinality constraints [5]. Recently, Sviridenko et al. [25] improved this approximation to $1 - c/e$ with variants of the continuous greedy and local search algorithms. Submodular optimization and curvature have also been studied for more general constraints [27, 15] and submodular minimization [16]. The curvature assumption has applications in problems such as maximum entropy sampling [25], column-subset selection [25], and submodular welfare [27].

## 2 Optimization from samples

We precisely define the framework of optimization from samples. A sample $(S, f(S))$ of function $f(\cdot)$ is a set and its value. As with the PMAC-learning framework, the samples $(S_i, f(S_i))$ are such that the sets $S_i$ are drawn i.i.d. from a distribution $\mathcal{D}$. As with the standard optimization framework, the goal is to return a set $S$ satisfying some constraint $\mathcal{M} \subseteq 2^N$ such that $f(S)$ is an $\alpha$-approximation to the optimal solution $f(S^\star)$ with $S^\star \in \mathcal{M}$.

A class of functions $\mathcal{F}$ is $\alpha$-*optimizable from samples* under constraint $\mathcal{M}$ and over distribution $\mathcal{D}$ if for all functions $f(\cdot) \in \mathcal{F}$ there exists an algorithm which, given polynomially many samples $(S_i, f(S_i))$, returns with high probability over the samples a set $S \in \mathcal{M}$ such that

$$f(S) \geq \alpha \cdot \max_{T \in \mathcal{M}} f(T).$$

In the unconstrained case, a random set achieves a $1/4$-approximation for general (not necessarily monotone) submodular functions [7]. We focus on the constrained case and consider a simple cardinality constraint $\mathcal{M}$, i.e., $\mathcal{M} = \{S : |S| \leq k\}$. To avoid trivialities in the framework, it is important to fix a distribution $\mathcal{D}$. We consider the distribution $\mathcal{D}$ to be the uniform distribution over all feasible sets, i.e., all sets of size at most $k$.

We are interested in functions that are both learnable and optimizable. It is already known that there exists classes of functions, such as coverage and submodular, that are both learnable and optimizable but not optimizable from samples for $\mathcal{M}$ and $\mathcal{D}$ defined above. This paper studies optimization from samples under some additional assumption: curvature. We assume that the curvature $c$ of the function is known to the algorithm designer. In the appendix, we show an impossibility result for learning the curvature of a function from samples.

## 3 An optimal algorithm

We design a $(1-c)/(1+c-c^2)-o(1)$-optimization from samples algorithm for monotone submodular functions with curvature $c$. In the next section, we show that this approximation ratio is tight. The main contribution is improving over the $(1-c)^2 - o(1)$ approximation algorithm from [4] to obtain a tight bound on the approximation.

**The algorithm.** Algorithm 1 first estimates the expected marginal contribution of each element $e_i$ to a uniformly random set of size $k-1$, which we denote by $R$ for the remaining of this section. These expected marginal contributions $\mathbf{E}_R[f_R(e_i)]$ are estimated with $\hat{v}_i$. The estimates $\hat{v}_i$ are the differences between the average value $\mathrm{avg}(\mathcal{S}_{k,i}) := (\sum_{T \in \mathcal{S}_{k,i}} f(T))/|\mathcal{S}_{k,i}|$ of the collection $\mathcal{S}_{k,i}$ of samples of size $k$ containing $e_i$ and the average value of the collection $\mathcal{S}_{k-1,i-1}$ of samples of size $k-1$ not containing $e_i$. We then wish to return the best set between the random set $R$ and the set $S$ consisting of the $k$ elements with the largest estimates $\hat{v}_i$. Since we do not know the value of $S$, we lower bound it with $\hat{v}_S$ using the curvature property. We estimate the expected value $\mathbf{E}_R[f(R)]$ of $R$ with $\hat{v}_R$, which is the average value of the collection $\mathcal{S}_{k-1}$ of all samples of size $k-1$. Finally, we compare the values of $S$ and $R$ using $\hat{v}_S$ and $\hat{v}_R$ to return the best of these two sets.

---

**Algorithm 1** A tight $(1-c)/(1+c-c^2)-o(1)$-optimization from samples algorithm for monotone submodular functions with curvature $c$

**Input:** $\mathcal{S} = \{S_i \ : \ (S_i, f(S_i)) \text{ is a sample}\}$
1: $\hat{v}_i \leftarrow \mathrm{avg}(\mathcal{S}_{k,i}) - \mathrm{avg}(\mathcal{S}_{k-1,i-1})$
2: $S \leftarrow \mathrm{argmax}_{|T|=k} \sum_{i \in T} \hat{v}_i$
3: $\hat{v}_S \leftarrow (1-c) \sum_{e_i \in S} \hat{v}_i$                    a lower bound on the value of $f(S)$
4: $\hat{v}_R \leftarrow \mathrm{avg}(\mathcal{S}_{k-1})$                    an estimate of the value of a random set $R$
5: **if** $\hat{v}_S \geq \hat{v}_R$ **then**
6:     **return** $S$
7: **else**
8:     **return** $R$
9: **end if**

---

**The analysis.** Without loss of generality, let $S = \{e_1, \ldots, e_k\}$ be the set defined in Line 2 of the algorithm and define $S_i$ to be the first $i$ elements in $S$, i.e., $S_i := \{e_1, \ldots, e_i\}$. Similarly, for the optimal solution $S^\star$, we have $S^\star = \{e_1^\star, \ldots, e_k^\star\}$ and $S_i^\star := \{e_1^\star, \ldots, e_i^\star\}$. We abuse notation and denote by $f(R)$ and $f_R(e)$ the expected values $\mathbf{E}_R[f(R)]$ and $\mathbf{E}_R[f_R(e)]$ where the randomization is over the random set $R$ of size $k-1$.

At a high level, the curvature property is used to bound the loss from $f(S)$ to $\sum_{i \leq k} f_R(e_i)$ and from $\sum_{i \leq k} f_R(e_i^\star)$ to $f(S^\star)$. By the algorithm, $\sum_{i \leq k} f_R(e_i)$ is greater than $\sum_{i \leq k} f_R(e_i^\star)$. When bounding the loss from $\sum_{i \leq k} f_R(e_i^\star)$ to $f(S^\star)$, a key observation is that if this loss is large, then it must be the case that $R$ has a high expected value. This observation is formalized in our analysis by bounding this loss in terms of $f(R)$ and motivates Algorithm 1 returning the best of $R$ and $S$. Lemma 1 is the main part of the analysis and gives an approximation for $S$. The approximation guarantee for Algorithm 1 (formalized as Theorem 1) follows by finding the worst-case ratios of $f(R)$ and $f(S)$.

**Lemma 1.** *Let $S$ be the set defined in Algorithm 1 and $f(\cdot)$ be a monotone submodular function with curvature $c$, then*

$$f(S) \geq (1-o(1))\hat{v}_S \geq \left((1-c)\left(1 - c \cdot \frac{f(R)}{f(S^\star)}\right) - o(1)\right) f(S^\star).$$

*Proof.* First, observe that

$$f(S) = \sum_{i \le k} f_{S_{i-1}}(e_i) \ge (1-c) \sum_{i \le k} f(e_i) \ge (1-c) \sum_{i \le k} f_R(e_i)$$

where the first inequality is by curvature and the second is by monotonicity. We now claim that w.h.p. and with a sufficiently large polynomial number of samples the estimates of the marginal contribution of an element are precise,

$$f_R(e_i) + \frac{f(S^\star)}{n^2} \ge \hat{v}_i \ge f_R(e_i) - \frac{f(S^\star)}{n^2}$$

and defer the proof to the appendix. Thus $f(S) \ge (1-c) \sum_{i \le k} \hat{v}_i - f(S^\star)/n \ge \hat{v}_S - f(S^\star)/n$. Next, by the definition of $S$ in the algorithm, we get

$$\frac{\hat{v}_S}{1-c} = \sum_{i \le k} \hat{v}_i \ge \sum_{i \le k} \hat{v}_i^\star \ge \sum_{i \le k} f_R(e_i^\star) - \frac{f(S^\star)}{n}.$$

It is possible to obtain a $1-c$ loss between $\sum_{i \le k} f_R(e_i^\star)$ and $f(S^\star)$ with a similar argument as in the first part. The key idea to improve this loss is to use the curvature property on the elements in $R$ instead of on the elements $e_i^\star \in S^\star$. By curvature, we have that $f_{S^\star}(R) \ge (1-c)f(R)$. We now wish to relate $f_{S^\star}(R)$ and $\sum_{i \le k} f_R(e_i^\star)$. Note that $f(S^\star) + f_{S^\star}(R) = f(R \cup S^\star) = f(R) + f_R(S^\star)$ by the definition of marginal contribution and $\sum_{i \le k} f_R(e_i^\star) \ge f_R(S^\star)$ by submodularity. We get $\sum_{i \le k} f_R(e_i^\star) \ge f(S^\star) + f_{S^\star}(R) - f(R)$ by combining the previous equation and inequality. By the previous curvature observation, we conclude that

$$\sum_{i \le k} f_R(e_i^\star) \ge f(S^\star) + (1-c)f(R) - f(R) = \left(1 - c \cdot \frac{f(R)}{f(S^\star)}\right) f(S^\star).$$

$\square$

Combining Lemma 1 and the fact that we obtain value at least $\max\{f(R), (1-c) \sum_{i=1}^{k} \hat{v}_i\}$, we obtain the main result of this section.

**Theorem 1.** *Let $f(\cdot)$ be a monotone submodular function with curvature c. Then Algorithm 1 is a $(1-c)/(1+c-c^2) - o(1)$ optimization from samples algorithm.*

*Proof.* In the appendix, we show that the estimate $\hat{v}_R$ of $f(R)$ is precise, the estimate is such that $f(R) + f(S^\star)/n^2 \ge \hat{v}_R \ge f(R) - f(S^\star)/n^2$. In addition, by the first inequality in Lemma 1, $f(S) \ge (1-o(1))\hat{v}_S$. So by the algorithm and the second inequality in Lemma 1, the approximation obtained by the set returned is at least

$$(1-o(1)) \cdot \max\left\{\frac{f(R)}{f(S^\star)}, \frac{\hat{v}_S}{f(S^\star)}\right\} \ge (1-o(1)) \cdot \max\left\{\frac{f(R)}{f(S^\star)}, (1-c)\left(1 - c \cdot \frac{f(R)}{f(S^\star)}\right)\right\}.$$

Let $x := f(R)/f(S^*)$, the best of $f(R)/f(S^\star)$ and $(1-c)(1 - c \cdot f(R)/f(S^\star)) - o(1)$ is minimized when $x = (1-c)(1-cx)$, or when $x = (1-c)/(1+c-c^2)$. Thus, the approximation obtained is at least $(1-c)/(1+c-c^2) - o(1)$. $\square$

## 4 Hardness

We show that the approximation obtained by Algorithm 1 is tight. For every $c < 1$, there exists monotone submodular functions that cannot be $(1-c)/(1+c-c^2)$-optimized from samples. This impossibility result is information theoretic, we show that with high probability the samples do not contain the right information to obtain a better approximation.

**Technical overview.** To obtain a tight bound, all the losses from Algorithm 1 must be tight. We need to obtain a $1 - cf(R)/f(S^\star)$ gap between the contribution of optimal elements to a random set $\sum_{i \le k} f_R(e_i^\star)$ and the value $f(S^\star)$. This gap implies that as a set grows with additional random elements, the contribution of optimal elements must decrease. The main difficulty is in obtaining this decrease while maintaining random sets of small value, submodularity, and the curvature.

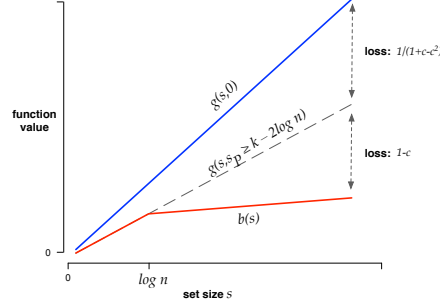

Figure 1: The symmetric functions $g(s_G, s_P)$ and $b(s_B)$.

The ground set of elements is partitioned into three parts: the good elements $G$, the bad elements $B$, and the poor elements $P$. In relation to the analysis of the algorithm, the optimal solution $S^\star$ is $G$, the set $S$ consists mostly of elements in $B$, and a random set consists mostly of elements in $P$. The values of the good, bad, and poor elements are given by the good, bad, and poor functions $g(\cdot)$, $b(\cdot)$, and $p(\cdot)$ to be later defined and the functions $f(\cdot)$ we construct for the impossibility result are:

$$f^G(S) := g(S \cap G, S \cap P) + b(S \cap B) + p(S \cap P).$$

The value of the good function is also dependent on the poor elements to obtain the decrease in marginal contribution of good elements mentioned above. The proof of the hardness result (Theorem 2) starts with concentration bounds in Lemma 2 to show that w.h.p. every sample contains a small number of good and bad elements and a large number of poor elements. Using these concentration bounds, Lemma 3 gives two conditions on the functions $g(\cdot)$, $b(\cdot)$, and $p(\cdot)$ to obtain the desired result. Informally, the first condition is that good and bad elements cannot be distinguished while the second is that $G$ has larger value than a set with a small number of good elements. We then construct these functions and show that they satisfy the two conditions in Lemma 4. Finally, Lemma 5 shows that $f(\cdot)$ is monotone submodular with curvature $c$.

**Theorem 2.** *For every $c < 1$, there exists a hypothesis class of monotone submodular functions with curvature $c$ that is not $(1-c)/(1+c-c^2) + o(1)$ optimizable from samples.*

The remaining of this section is devoted to the proof of Theorem 2. Let $\epsilon > 0$ be some small constant. The set of poor elements $P$ is fixed and has size $n - n^{2/3 - \epsilon}$. The good elements $G$ are then a uniformly random subset of $P^C$ of size $k := n^{1/3}$, the remaining elements $B$ are the bad elements. The following concentration bound is used to show that elements in $G$ and $B$ cannot be distinguished. The proof is deferred to the appendix.

**Lemma 2.** *All samples $S$ are such that $|S \cap (G \cup B)| \leq \log n$ and $|S \cap P| \geq k - 2\log n$ w.h.p..*

We now give two conditions on the good, bad, and poor functions to obtain an impossibility result based on the above concentration bounds. The first condition ensures that good and bad elements cannot be distinguished. The second condition quantifies the gap between the value of $k$ good elements and a set with a small number of good elements. We denote by $s_G$ the number of good elements in a set $S$, i.e., $s_G := |S \cap G|$ and define similarly $s_B$ and $s_P$. The good, bad, and, poor functions are symmetric, meaning they each have equal value over sets of equal size, and we abuse the notation with $g(s_G, s_P) = g(S \cap G, S \cap P)$ and similarly for $b(s_B)$ and $p(s_P)$. Figure 1 is a simplified illustration of these two conditions.

**Lemma 3.** *Consider sets $S$ and $S'$, and assume $g(\cdot)$, $b(\cdot)$, and $p(\cdot)$ are such that*

1. *$g(s_G, s_P) + b(s_B) = g(s'_G, s'_P) + b(s'_B)$ if*
    - *$s_G + s_B = s'_G + s'_B \leq \log n$ and $s_P, s'_P \geq k - 2\log n$,*

2. *$g(s_G, s_P) + b(s_B) + p(s_P) < \alpha \cdot g(k, 0)$ if*
    - *$s_G \leq n^\epsilon$ and $s_G + s_B + s_P \leq k$*

*then the hypothesis class of functions $\mathcal{F} = \{f^G(\cdot) \,:\, G \subseteq P^C, |G| = k\}$ is not $\alpha$-optimizable from samples.*

*Proof.* By Lemma 2, for any two samples $S$ and $S'$, $s_G + s_B \leq \log n$, $s'_G + s'_B \leq \log n$ and $s_P, s'_P \geq k - 2\log n$ with high probability. If $s_G + s_B = s'_G + s'_B$, then by the first assumption, $g(s_G, s_P) + b(s_B) = g(s'_G, s'_P) + b(s'_B)$. Recall that $G$ is a uniformly random subset of the fixed set $P^C$ and that $B$ consists of the remaining elements in $P^C$. Thus, w.h.p., the value $f^G(S)$ of all samples $S$ is *independent* of which random subset $G$ is. In other words, no algorithm can distinguish good elements from bad elements with polynomially many samples. Let $T$ be the set returned by the algorithm. Since any decision of the algorithm is independent from $G$, the expected number of good elements in $T$ is $t_G \leq k \cdot |G|/|G \cup B| = k^2/n^{2/3-\epsilon} = n^\epsilon$. Thus,

$$\mathbf{E}_G\big[f^G(T)\big] = g(t_G, t_P) + b(t_B) + p(t_P) \leq g(n^\epsilon, t_P) + b(t_B) + p(t_P) < \alpha \cdot g(k, 0)$$

where the first inequality is by the submodularity and monotonicity properties of the good elements $G$ for $f^G(\cdot)$ and the second inequality is by the second condition of the lemma. By expectations, the set $S$ returned by the algorithm is therefore not an $\alpha$-approximation to the solution $G$ for at least one function $f^G(\cdot) \in \mathcal{F}$ and $\mathcal{F}$ is not $\alpha$-optimizable from samples. $\square$

**Constructing** $g(\cdot), b(\cdot), p(\cdot)$. The goal is now to construct $g(\cdot)$, $b(\cdot)$ and $p(\cdot)$ that satisfy the above conditions. We start with the good and bad function:

$$g(s_G, s_P) = \begin{cases} s_G \cdot \left(1 - \left(1 - \frac{1}{1+c-c^2}\right) \cdot s_P \cdot \frac{1}{k-2\log n}\right) & \text{if } s_p \leq k - 2\log n \\ s_G \cdot \frac{1}{1+c-c^2} & \text{otherwise} \end{cases}$$

$$b(s_B) = \begin{cases} s_B \cdot \frac{1}{1+c-c^2} & \text{if } s_B \leq \log n \\ (s_B - \log n) \cdot \frac{1-c}{1+c-c^2} + \log n \cdot \frac{1}{1+c-c^2} & \text{otherwise} \end{cases}$$

These functions exactly exhibit the losses from the analysis of the algorithm in the case where the algorithm returns bad elements. As illustrated in Figure 1, there is a $1 - c$ loss between the contribution $1/(1+c-c^2)$ of a bad element to a random set and its contribution $(1-c)/(1+c-c^2)$ to a set with at least $\log n$ bad elements. There is also a $1/(1+c-c^2)$ loss between the contribution $1$ of a good element to a set with no poor elements and its contribution $1/(1+c-c^2)$ to a random set. We add a function $p(s_P)$ to $f^G(\cdot)$ so that it is monotone increasing when adding poor elements.

$$p(s_P) = \begin{cases} s_p \cdot \frac{1-c}{1+c-c^2} \cdot \frac{k}{k-2\log n} & \text{if } s_P \leq k - 2\log n \\ \left((s_p - (k - 2\log n))\frac{(1-c)^2}{1+c-c^2} + (k - 2\log n)\frac{1-c}{1+c-c^2}\right)\frac{k}{k-2\log n} & \text{otherwise} \end{cases}$$

The next two lemmas show that theses function satisfy Lemma 3 and that $f^G(\cdot)$ is monotone submodular with curvature $c$, which concludes the proof of Theorem 2.

**Lemma 4.** *The functions* $g(\cdot)$*,* $b(\cdot)$*, and* $p(\cdot)$ *defined above satisfy the conditions of Lemma 3 with* $\alpha = (1 - c)/(1 + c - c^2) + o(1)$.

*Proof.* We start with the first condition. Assume $s_G + s_B = s'_G + s'_B \leq \log n$ and $s_P, s'_P \geq k - 2\log n$. Then,

$$g(s_G, s_P) + b(s_B) = (s_G + s_B) \cdot \frac{1}{1+c-c^2} = (s'_G + s'_B) \cdot \frac{1}{1+c-c^2} = g(s'_G, s'_P) + b(s'_B).$$

For the second condition, assume $s_G \leq n^\epsilon$ and $s_G + s_B + s_P \leq k$. It is without loss to assume that $s_B + s_P \geq k - n^\epsilon$, then

$$f^G(S) \leq (1 + o(1)) \cdot (s_B + s_P) \cdot \frac{1-c}{1+c-c^2} \leq k \cdot \left(\frac{1-c}{1+c-c^2} + o(1)\right).$$

We conclude by noting that $g(k, 0) = k$. $\square$

**Lemma 5.** *The function* $f^G(\cdot)$ *is a monotone submodular function with curvature* $c$.

*Proof.* We show that the marginal contributions are positive (monotonicity), decreasing (submodularity), but not by more than a $1 - c$ factor (curvature), i.e., that $f_S(e) \geq f_T(e) \geq (1-c)f_S(e) \geq 0$ for all $S \subseteq T$ and $e \notin T$. Let $e$ be a **good element**, then

$$f^G_S(e) = \begin{cases} \left(1 - \left(1 - \frac{1}{1+c-c^2}\right) \cdot s_P \cdot \frac{1}{k-2\log n}\right) & \text{if } s_p \leq k - 2\log n \\ \frac{1}{1+c-c^2} & \text{otherwise.} \end{cases}$$

Since $s_P \leq t_P$ for $S \subseteq T$, we obtain $f_S(e) \geq f_T(e) \geq 0$. It is also easy to see that we get $f_T(e) \geq \frac{1}{1+c-c^2} \geq (1-c) \geq (1-c)f_S(e)$. For **bad elements**,

$$f_S^G(e) = \begin{cases} \frac{1}{1+c-c^2} & \text{if } s_B \leq \log n \\ \frac{1-c}{1+c-c^2} & \text{otherwise.} \end{cases}$$

Thus, $f_S(e) \geq f_T(e) \geq (1-c)f_S(e) \geq 0$ for all $S \subseteq T$ and $e \notin T$. Finally, for **poor elements**,

$$f_S^G(e) = \begin{cases} -\left(1 - \frac{1}{1+c-c^2}\right) \cdot s_G \cdot \frac{1}{k-2\log n} + \frac{1-c}{1+c-c^2} \cdot \frac{k}{k-2\log n} & \text{if } s_P \leq k - 2\log n \\ \frac{(1-c)^2}{1+c-c^2}\frac{k}{k-2\log n} & \text{otherwise.} \end{cases}$$

Since $s_G \leq k$,

$$\frac{1-c}{1+c-c^2} \cdot \frac{k}{k-2\log n} \geq f_S^G(e) \geq \frac{(1-c)^2}{1+c-c^2}\frac{k}{k-2\log n}.$$

Consider $S \subseteq T$, then $s_G \leq t_G$, and $f_S(e) \geq f_T(e) \geq (1-c)f_S(e) \geq 0$. $\qquad\qquad\square$

## 5 Experiments

We perform simulations on simple synthetic functions. These experiments are meant to complement the theoretical analysis by conveying some interpretations of the bounds obtained. The synthetic functions are a simplification of the construction for the impossibility result. The motivation for these functions is to obtain hard instances that are challenging for the algorithm. More precisely, the function considered is

$$f(S) = \begin{cases} |S \cap (G \cup B)| & \text{if } |S \cap B| \leq 10 \\ |S \cap G| + |S \cap B| \cdot (1-c) - 10c \end{cases}$$

otherwise, where $G$ and $B$ are fixed sets of size $10^2$ and $10^3$ respectively. The ground set $N$ contains $10^5$ elements. It is easy to verify that $f(\cdot)$ has curvature $c$. This function is hard to optimize since the elements in $G$ and $B$ cannot be distinguished from samples.

We consider several benchmarks. The first is the value obtained by the learn then optimize approach where we first learn the function and then optimize the learned function. Equivalently, this is a random set of size $k$, since the learned function is a constant with the algorithm from [3]. We also compare our algorithm to the value of the best sample observed. The solution returned by the greedy algorithm is an upper bound and is a solution obtainable only in the full information setting. The results are summarized in Figure 2 and 3. In Figure 2, the value of greedy, best sample, and random set do not change for different curvatures $c$ since w.h.p. they pick at most 10 elements from $B$. For curvature $c = 0$, when the function is modular, our algorithm performs as well as the greedy algorithm, which is optimal. As the

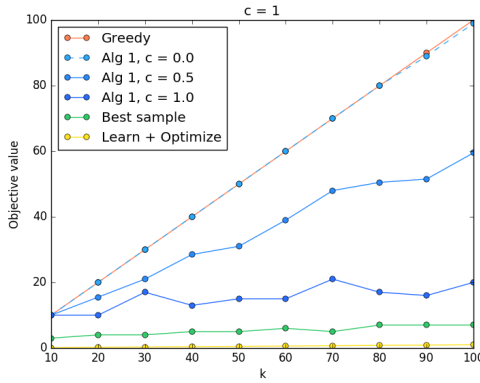

Figure 2: The objective $f(\cdot)$ as a function of the cardinality constraint $k$.

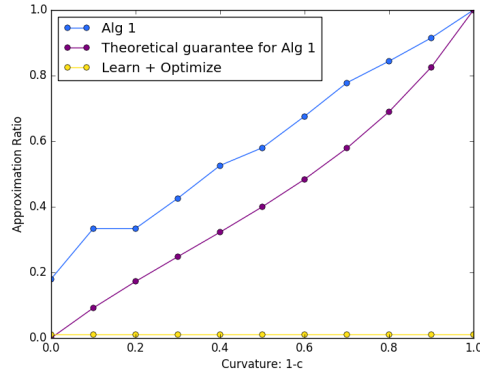

Figure 3: The approximation as a function of the curvature $1 - c$ when $k = 100$.

curvature increases, the solution obtained by our algorithm worsens, but still significantly outperforms the best sample and a random set. The power of our algorithm is that it is capable to distinguish elements in $G \cup B$ from the other elements.

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
