[Supplementary Material · samplesCurvature.pdf]

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

# Appendix

We restate the statement of the missing results for convenience.

## A Missing Proof from Section 3

**Claim.** *Let $f$ be a monotone submodular function. Then, with a sufficiently large polynomial number of samples, the estimations $\hat{v}_i$ and $\hat{v}_R$ are $f(S^\star)/n^2$-close to $f_R(e_i)$ and $f(R)$ with high probability, i.e.,*

$$f_R(e_i) + \frac{f(S^\star)}{n^2} \geq \hat{v}_i \geq f_R(e_i) - \frac{f(S^\star)}{n^2},$$

*and*

$$f(R) + \frac{f(S^\star)}{n^2} \geq \hat{v}_R \geq f(R) - \frac{f(S^\star)}{n^2}.$$

*Proof.* We assume that $k \leq n/2$ (otherwise, a random subset of size $k$ is a $1/2$-approximation). The size of a sample which is the most likely is $k$, so the probability that a sample is of size $k$ is at least $2/n$. Since $\binom{n}{k-1} \geq \binom{n}{k}/n$, the probability that a sample is of size $k-1$ is at least $2/n^2$. A given element $i$ has probability at least $1/n$ of being in a sample and probability at least $1/2$ of not being in a sample. Therefore, to observe at least $n^5$ samples of size $k$ which contain $i$ and at least $n^5$ samples of size $k-1$ which do not contain $i$, $n^8$ samples are sufficient with high probability. Since $f(S) \leq f(S^\star)$ for all samples $S$, by Hoeffding's inequality,

$$\Pr\left( \left| \mathrm{avg}(\mathcal{S}_{k,i}) - \mathbf{E}_{S\,:\,|S|=k, i \in S}[f(S)] \right| \geq \frac{f(S^\star)}{2n^2} \right) \leq 2e^{-2n^5(f(S^\star)/2n^2)^2/f(S^\star)^2} \leq 2e^{-n/2}.$$

similarly,

$$\Pr\left( \left| \mathrm{avg}(\mathcal{S}_{k-1,i-1}) - \mathbf{E}_{S\,:\,|S|=k-1, i \notin S}[f(S)] \right| \geq \frac{f(S^\star)}{2n^2} \right) \leq 2e^{-n/2}$$

and

$$\Pr\left( \left| \mathrm{avg}(\mathcal{S}_{k-1}) - \mathbf{E}_{S\,:\,|S|=k-1}[f(S)] \right| \geq \frac{f(S^\star)}{2n^2} \right) \leq 2e^{-n/2}.$$

Since $\hat{v}_i = \mathrm{avg}(\mathcal{S}_{k,i}) - \mathrm{avg}(\mathcal{S}_{k-1,i-1})$, $f_R(e_i) = \mathbf{E}_{S\,:\,|S|=k, i \in S}[f(S)] - \mathbf{E}_{S\,:\,|S|=k-1, i \notin S}[f(S)]$, $\hat{v}_R = \mathrm{avg}(\mathcal{S}_{k-1})$, and $f(R) = \mathbf{E}_{S\,:\,|S|=k-1}[f(S)]$, the claim holds with high probability. $\qquad\square$

## B Missing Proof from Section 4

**Lemma.** *With high probability, all samples $S$ are such that*

- $|S \cap (G \cup B)| \leq \log n$, *and*
- $|S \cap P| \geq k - 2\log n$.

*Proof.* We start by showing the first part. Consider a subset $L$ of $G \cup B$ of size $\log n$. We first bound the probability that $L$ is a subset of a sample $S$,

$$\Pr(L \subseteq S) \leq \prod_{e \in L} \Pr(e \in S) \leq \prod_{e \in L} \frac{k}{n} = \left( \frac{k}{n} \right)^{\log n}.$$

We then bound the probability that $|S \cap (G \cup B)| > \log n$ with a union bound over the events that a set $L$ is a subset of $S$, for all subsets $L$ of $T$ of size $\log n$:

$$\begin{aligned}
\Pr(|S \cap (G \cup B)| > \log n) &\leq \sum_{L \subseteq G \cup B\,:\,|L|=\log n} \Pr(L \subseteq S) \\
&\leq \binom{|G \cup B|}{\log n} \cdot \left( \frac{k}{n} \right)^{\log n} \\
&\leq \left( \frac{k \cdot |G \cup B|}{n} \right)^{\log n} \\
&\leq n^{-\epsilon \log n}.
\end{aligned}$$

We now show that a sample is of size at least $k - \log n$ w.h.p., which combined with the first part of the lemma, implies the second part. We lower bound the ratio of the number of sets of size $k$ to the number of sets of size at most $k - \log n$:

$$\frac{\binom{n}{k}}{\sum_{i=0}^{k-\log n} \binom{n}{i}} \geq \frac{\binom{n}{k}}{k \cdot \binom{n}{k-\log n}}$$

$$= \frac{1}{k} \prod_{i=0}^{\log n - 1} \frac{\binom{n}{k-i}}{\binom{n}{k-i-1}}$$

$$= \frac{1}{k} \prod_{i=0}^{\log n - 1} \frac{n - k + i - 1}{k - i}$$

$$\geq \frac{1}{k} \left(\frac{n}{k}\right)^{\log n} (1 - o(1))$$

$\square$

## C  Impossibility result for learning curvature

**Claim.** *There exists two functions $f_0(\cdot)$ and $f_1(\cdot)$ that have curvature $0$ and $1$ respectively and that cannot be distinguished from samples.*

*Proof.* Let $G$ be a set of size $k = n^{1/3}$. Then let

$$f_0(S) = |G \cap S| \quad \text{and} \quad f_1(S) = \min(|G \cap S|, \log n).$$

By Lemma 2, all samples contain at most $\log n$ elements form $G$ with high probability. Thus $f_1(S) = |G \cap S| = f_0(S)$ for all samples $S$ with high probability and these functions are not distinguishable from samples. It is easy to verify that these functions have curvature $0$ and $1$. $\square$