[Reviews · NeurIPS 2016]

Reviewer 1

Summary

This paper considers the problem of maximizing a monotone submodular function under cardinality constraints given the valuation of polynomially many sets chosen uniformly from the feasible region. While there are previous lower bounds for performing this task in general they show how to obtain improved results when the function as bounded curvature. Formally they show that for any submodular function f with curvature c it is possible to obtain a (1 – c)/(1 + c – c^2) – o(1) approximation. Moreover, they show that this is tight by proving that for every c there is a submodular function with curvature c for which it is impossible to obtain a better than (1 – c)/(1 + c – c^2) approximation ratio in general using a polynomial number of samples.

Qualitative Assessment

The paper is clear, well, written, and provides a tight characterization of a fairly natural problem regarding submodular function maximization. Moreover, they motivate their problem well by contrasting their result with results on learning submodular function and highlighting that learning doesn’t necessary explain whether or not the function is optimizable. Finally, since the algorithm they provide and analyze is not necessarily the simplest one would imagine, their paper is possibly motivating new algorithms for submodular maximization in certain contexts. While a little more discussion of how the lower bound in this paper compares to previous work and whether or not certain parts of the algorithm are essential (e.g. can the random R be replaced with sets seen) would benefit the paper, overall it is a solid submission. Detailed Comments • Line 6: maybe the words with high probability or something about the success probability should be added? • Line 10: maybe worth saying how many query can rule out? Is it exponential or something between exponential and polynomial? • Line 62: what is meant by impossible? No multiplicative approximation ratio? • Line 68: there should be some citations right after saying that curvature has been heavily studied • Line 115-120: are these bounds tight? The best lower bound on the problem in the full information case should be stated for completeness. • Line 152: putting the definition or construction of R into the algorithm might help the clarify somewhat

Confidence in this Review

2-Confident (read it all; understood it all reasonably well)


Reviewer 2

Summary

The paper considers the problem of maximizing a submodular function with a cardinality constraint, under the model in which the algorithm only sees the function value at random points of the input space. For monotone submodular functions of bounded curvature 'c', the paper presents matching upper and lower bounds on the approximation ratio. The lower bound is an information theoretic one (as opposed to a hardness), which is nice.

Qualitative Assessment

The model considered in the paper, obtaining function values at random feasible points of the input space, seems realistic enough, and the algorithm is simple to state. The analysis is elegant and short, and it's nice that such a simple algorithm is optimal. The lower bound technique is also clean, and the paper gets matching bounds. The only negative is the somewhat limited applicability. Bounded curvature submodular functions are considered "very easy", and it's almost immediate to obtain a constant approximation. Figuring out the precise constant is the main contribution.

Confidence in this Review

2-Confident (read it all; understood it all reasonably well)


Reviewer 3

Summary

The paper studies submodular maximization subject to a cardinality constraint of k and the access to the function is restricted to uniform samples. While this task is impossible in general, the paper considers the case of bounded curvature and shows matching upper and lower bounds for the approximation factor of (1-c)/(1+c-c^2), where c is the curvature. The key insight is to estimate the marginal gain of each element with respect to a random set of size k-1. The solution is either the set of k elements with maximum estimated marginal gains over random sets or a random set.

Qualitative Assessment

The result is interesting as approximation from samples is not possible in general but the author identified the case of bounded curvature as a common scenario where good approximation is possible and derive best possible approximation bound for this setting. The lower bound example is quite intricate and highlight the difficulty with the uniform sample model. The experiment could be better, is there no real dataset that makes sense for this problem?

Confidence in this Review

2-Confident (read it all; understood it all reasonably well)


Reviewer 4

Summary

Optimization from samples of monotone submodular functions with bounded curvature is studied. The authors provide and prove hard bounds on the approximation ratio for this problem. Specifically, using a probabilistic argument, they show that the approximation bounds are tight. They focus on the constrained case and consider the problem of maximizing the function with a cardinality constraint and provide a simple algorithm to solve the problem. Simulation study suggests that the algorithm is better than the best sample and a random set.

Qualitative Assessment

The paper is overall well written, in particular: section 1. sets up the problem well; intuition for the proofs were helpful in checking them and I think it should be accessible to readers with some basic understanding of submodular optimization literature. The references given by the authors are standard, good and popular which is good. In particular, I liked the hardness section where the idea is to classify elements of the ground set into two categories: good, bad and poor. Poor elements are only useful for making the constructed function monotone submodular whereas the good and bad are elements that carry very less "information". I am not sure if information is the right word here since usually it is related to entropy functions, it will be nice if the authors can describe why they call it that way. One major concern that I have is the simulation experiments. In general, I think that a single random set is not fair for comparison, if you are going to compare it with randomized selection, I think many draws should be conducted for evaluation (similar to boot strapping) and a confidence interval or bar would convey more than a single random draw. Secondly, while this seeme like a theoretical paper, nonetheless, I feel like it can/should be supplemented with some experiments using real datasets on problems that the authors mention in section 1. This is the main reason for my score for question 5, 7 (technical quality, Potential impact or usefulness). While the analysis works for the special case of cardinality constrained problems, any thoughts on how to adapt the analysis to other constraints like cover or knapsack constraints? (I suppose that this is not simple as it sounds and can be highly nontrivial?)

Confidence in this Review

2-Confident (read it all; understood it all reasonably well)


Reviewer 5

Summary

This paper shows that for any monotone submodular function with curvature c there is an approximation algorithm for maximization under cardinality constraints when polynomially-many samples are drawn from theuniform distribution over feasible sets.

Qualitative Assessment

General comments: -The paper is very well written. The presentation is easy to follow. -The theoretical results seem strong and sound. Both upper and lower bounds are proved. -On the machine learning application side, it is less clear how to use these results because of the polynomial sampling complexity. It would also be interesting to add numerical experiments on a real learning task (eg document ranking) and to compare the results to existing algorithms.

Confidence in this Review

1-Less confident (might not have understood significant parts)